# What did you choose just now? Chimpanzees' short-term retention of memories of their own behavior

Masaki Tomonaga[1] and Takaaki Kaneko[2]

[1] Primate Research Institute, Kyoto University, Kanrin, Inuyama, Aichi, Japan
[2] RIKEN Brain Science Institute, Wako, Saitama, Japan

## ABSTRACT

Many recent comparative studies have addressed "episodic" memory in nonhuman animals, suggesting that birds, rodents, great apes, and others can remember their own behavior after at least a half-day delay. By contrast, despite numerous studies regarding long-term memory, few comparable studies have been conducted on short-term retention for own behavior. In the current study, we addressed the following question: Do chimpanzees remember what they have just done? Four chimpanzees performed matching-to-sample and visual search tasks on a routine basis and were occasionally (every four sessions) given a "recognition" test immediately after their response during visual search trials. Even though these test trials were given very rarely, all four chimpanzees chose the stimulus they selected in the visual search trials immediately before the test trial significantly more frequently than they chose the stimulus they selected in another distractor trial. Subsequent experiments ruled out the possibility that preferences for the specific stimuli accounted for the recognition test results. Thus, chimpanzees remembered their own behavior even within a short-term interval. This type of memory may involve the transfer of episodic information from working memory to long-term episodic-like memory (i.e., an episodic buffer).

Corresponding author
Masaki Tomonaga,
tomonaga@pri.kyoto-u.ac.jp

Episodic memory is defined as long-term memory for personally experienced events (*Tulving, 1985*). Some researchers have suggested that episodic memory should contain information about "what" occurred, "when" it occurred, and "where" it occurred (e.g., *Clayton & Dickinson, 1998*; *Skov-Rackette, Miller & Shettleworth, 2006*). Due to recent advances in comparative cognitive science, many species have been found to show some degree of this "episodic" memory (e.g., food-caching jays (*Clayton & Dickinson, 1998*), rhesus macaques (*Hampton, Hampstead & Murray, 2005*; *Hoffman, Beran & Washburn, 2009*), rats (*Crystal & Babb, 2008*), and dolphins (*Mercado et al., 1998*)). For example, great apes showed "episodic" memory (*Menzel, 1999*; *Schwartz, Hoffman & Evans, 2005*; *Martin-Ordas et al., 2010*). *Menzel (1999)* reported that one symbol-trained chimpanzee remembered where an object was hidden in an outdoor enclosure for 16 h.

However, controversy about whether these examples of long-term memory satisfy the criteria of human-like episodic memory persists (*Suddendorf & Busby, 2003*; *Tulving,*

2005). Thus, the term "episodic-like" memory is often used to describe this type of memory in nonhuman animals (e.g., *Clayton & Dickinson, 1998*). One reason for this is that many of the studies mentioned above seemingly lack components of "what", "when", and "where". For example, *Hampton, Hampstead & Murray (2005)* administered a memory task to rhesus monkeys that was functionally equivalent to those used by *Clayton & Dickinson (1998)*; however, unlike the latter's results with jays, *Hampton, Hampstead & Murray (2005)* found intact memory for the "what" and "where" of past events but not for the "when".

Most previous studies of episodic-like memory in nonhuman animals have focused intensively on the "long-term" retention of episodes, primarily because of the definition of episodic memory mentioned above. However, the other aspects of memory of past episodes should also be examined. One neglected area is the transition from working (or short-term) memory to long-term episodic-like memory. In terms of human working memory, *Baddeley (2000)* proposed the "episodic buffer", a link between the working and long-term memory systems. However, how the short-term retention of working memories is crystalized into long-term episodic-like memories in nonhuman animals remains unclear (cf., *Zhou, Hohmann & Crystal, 2012*). To address this issue, research regarding the short-term properties of episodic-like memory is necessary.

Another aspect of episodic-like memory is the retroactive retrieval of incidentally encoded information. Humans can recall past events that have not been actively encoded, and several researchers have recently become interested in this aspect of memory in nonhuman animals, conducting relevant experimental studies (dogs: *Fujita et al., 2012*; rats: *Zhou, Hohmann & Crystal, 2012*; pigeons: *Singer & Zentall, 2007*; *Zentall et al., 2001*; *Zentall, Singer & Stagner, 2008*).

Zentall and his colleagues (*Singer & Zentall, 2007*; *Zentall et al., 2001*; *Zentall, Singer & Stagner, 2008*) tested pigeons for episodic-like memory under the matching-to-sample paradigm. For example, pigeons were trained to match the left position to red and the right position to green (position–color matching) and to match yellow to vertical lines and blue to horizontal lines (color–line matching). After learning to perform these two types of matching tasks, they completed probe trials in which red and green choices were occasionally presented after participants chose the vertical or horizontal stimulus in the color–line trials. When pigeons selected the left key for vertical lines, they chose the red key and vice versa. Pigeons exhibited significantly better performance than would be expected by chance (71.6%) during such probe trials (*Singer & Zentall, 2007*). This result indicates that pigeons remember their own behavior even when they are unexpectedly prompted to do so. The same experimental paradigm was also used by *Zhou, Hohmann & Crystal (2012)*, who also reported that rats can respond correctly to unexpected probe trials during maze-learning tasks. Some researchers have emphasized this unexpectedness in the episodic memory system (*Zentall, Singer & Stagner, 2008*); however, pigeons underwent eight probe trials in each of 24 104-trial sessions in Singer and Zentall's study (*2007*). Given such a relatively large number of probe trials, pigeons might have learned the specific procedure for the probe trials or the probe trials might have become less unexpected.

Consistent with the approach adopted by Zentall et al. (*Singer & Zentall, 2007*; *Zentall et al., 2001*; *Zentall, Singer & Stagner, 2008*) and *Zhou, Hohmann & Crystal (2012)*, we also focused on chimpanzees' retroactive retrieval of incidentally encoded information. To this end, we tested the short-term retention ability of chimpanzees for their own behavior. Chimpanzee participants performed matching-to-sample and visual search trials. In the matching trials, the chimpanzees were initially shown a single sample stimulus on a touch-screen monitor and were required to touch it. After a delay interval, two choice stimuli, one of which was the same as the sample, were presented. In contrast, the search display, which contained one target and five distractor stimuli, was presented during the visual search trials, and the chimpanzees were required to touch an odd item among the uniform distractors. As a probe test, chimpanzees were occasionally presented with two stimuli after they selected the target in the visual search trial. One of the stimuli was the one that the subject had chosen from the search display, and the other differed from the stimuli in the previous search display. These probe trials were presented once per test session, and the test session appeared every four baseline sessions to maintain the rarity and unexpectedness of the probe trials. If the chimpanzees remembered their own choice just before the probe test, they could choose the "correct" stimulus during the probe trial.

## METHODS

### Participants

Four chimpanzees (*Pan troglodytes*) from the Primate Research Institute, Kyoto University (KUPRI), participated: Pendesa (female, 28 years old at the onset of the present experiment), Ayumu (male, 7 years old), Cleo (female, 7 years old), and Pal (female, 7 years old). As these chimpanzees have a long history of comparative cognitive experiments involving matching-to-sample and visual search tasks (*Fagot & Tomonaga, 1999*; *Goto, Imura & Tomonaga, 2012*; *Tomonaga, 2001*; *Tomonaga & Imura, 2010*; *Matsuzawa, Tomonaga & Tanaka, 2006*), it was not necessary to train them for these tasks before the present experiments. They live in a social group of 14 individuals in an environmentally enriched outdoor compound (770 m$^2$) connected to the experimental room by a tunnel.

### Ethical considerations

The care and use of the chimpanzees adhered to the 2nd edition of the Guide for the Care and Use of Laboratory Primates issued by KUPRI in 2002. The design of the current research was approved by the Animal Welfare and Animal Care Committee of KUPRI and by the Animal Research Committee of Kyoto University (Approval No. 07-1554). All procedures adhered to the Japanese Act on the Welfare and Management of Animals.

### Apparatus and stimuli

Experimental sessions were conducted in a booth (1.8 × 2.15 × 1.75 m) in an experimental room. A 17-inch LCD monitor (IO-Data, Tokyo, Japan, Model LCD-AD171F-T, 1280 × 1,024 pixels, pixel size: 0.264 × 0.264 mm) with a touch screen was placed on a wall of the booth (Fig. 1) at a viewing distance of approximately 40 cm. A food reward was

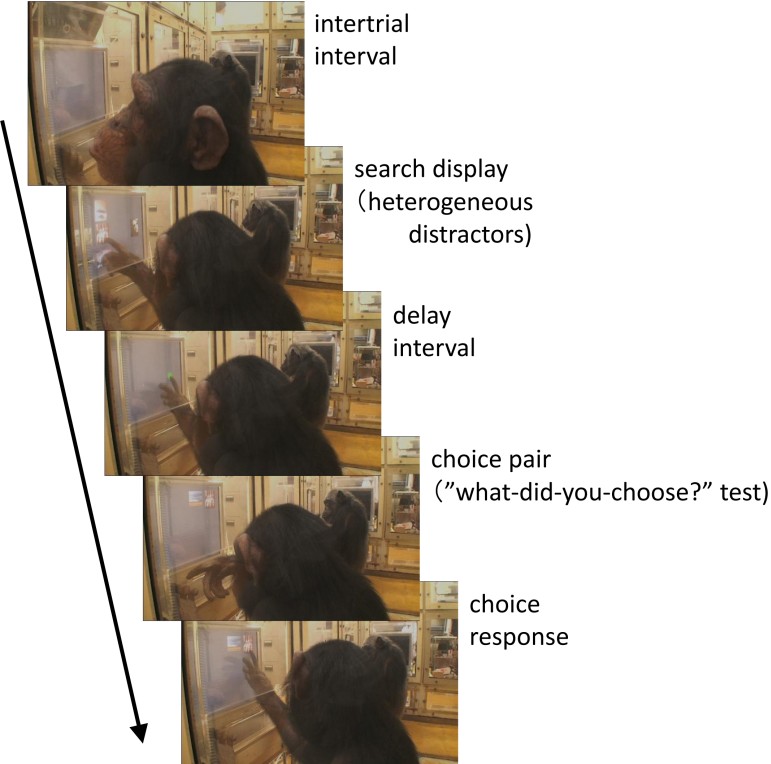

intertrial
interval

search display
(heterogeneous
distractors)

delay
interval

choice pair
("what-did-you-choose?" test)

choice
response

**Figure 1 Chimpanzee Cleo performing the "what-did-you-choose?" test.** Photo Credit: Masaki Tomonaga (Kyoto University).

delivered via a food tray connected to a universal feeder located under the monitor (Bio-Medica Ltd., Osaka, Japan, Model BUF-310). Experimental events and data collection were controlled by a PC using a customized program written in Microsoft Visual Basic 6.0©.

We prepared 2,500 color photographs (6.0 × 6.0 cm in size) depicting various kinds of objects and scenes as stimuli. We did not control any of the parameters of these pictures, such as the distribution of colors, average brightness, and so on. These stimuli were presented only once to each chimpanzee.

## Procedure

### *Matching-to-sample trials*

Figure 2A shows the flow of a matching-to-sample trial. In each trial, a blue square (3.0 × 3.0 cm) was presented at the bottom center of the monitor as a warning signal. When the chimpanzee touched the square, it disappeared, and a sample stimulus was presented at a random position on the monitor. Touching the sample stimulus terminated it, and a green circle (3.0 cm in diameter) was presented at a random location on the monitor. A single touch to this circle was followed by the presentation of the two choice stimuli. This interval was defined as a self-paced delay interval. Touching the choice stimulus that was the same as the sample delivered a food reward (a piece of apple or a raisin) followed by a chime. If the chimpanzee chose a stimulus that differed from

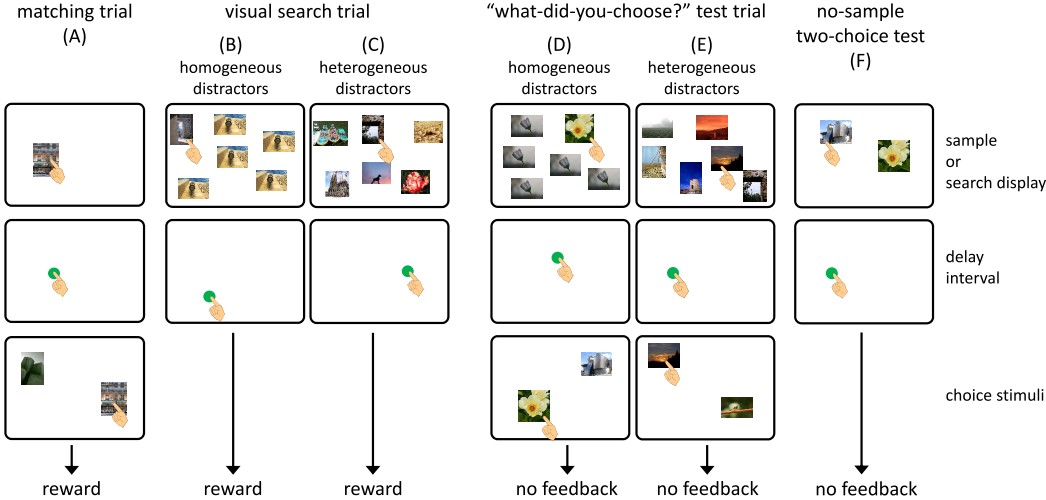

**Figure 2** **Flow of each type of trial in the present experiment.** Photo Credit: Masaki Tomonaga (Kyoto University).

the sample, this response was recorded as an error, and a buzzer sounded. The intertrial interval was 3 s.

## Visual search trials

As in the matching trials, each trial began with the presentation of a warning signal. After touching this, the search display was presented. We prepared two types of visual search trials. In the homogeneous-distractor trials, one target and five identical distractors were presented as a search display (Fig. 2B), whereas six different stimuli appeared in the heterogeneous-distractor trials (Fig. 2C). The chimpanzees were required to touch the target stimulus during the homogeneous-distractor trials, but any choice of stimulus was non-differentially reinforced during the heterogeneous-distractor trials. When the chimpanzees touched the target during the homogeneous-distractor trials and touched any stimulus during the heterogeneous-distractor trials, the search display disappeared, and a green circle was presented at a random location, as in the matching trials. Touching the green circle delivered a food reward accompanied by a chime. If the chimpanzees touched one of the distractors during the homogeneous trials (error response), the search display disappeared, and a buzzer sounded. The presentation of the green circle was intended to equalize the flow of the trials across the matching, visual search, and test trials (described later).

## "What-did-you-choose" test

The third type of trial was called a "what-did-you-choose" test trial (Figs. 2D and 2E). In this trial, two choice stimuli occasionally appeared after the self-paced delay interval following the visual search task. One of the two stimuli was the one that the subject chose from the search display, and the other was the one that was different from any stimulus in the previous search display. Any touch to a stimulus caused the termination of the stimuli, but was not followed by feedback or a food reward. It should be noted that if the

chimpanzees made an error during the homogeneous-distractor visual search component of the test trials, the subsequent two-choice component was cancelled.

Each baseline session consisted of 28 trials, in which the first four trials were matching trials. Among the remaining 24 visual search trials, 16 were homogeneous- and eight were heterogeneous-distractor trials. These trials were randomly presented. Test sessions were presented after every four sessions of the baseline trials, and either a homogeneous or heterogeneous test trial appeared as 29th trial. Each chimpanzee received 64 test sessions, i.e., one session per day, four or five times per week, yielding 64 test trials (32 for each type of test trial). All stimulus positions were randomly assigned across trials. The correct stimulus position was also randomized among trials.

## Control tests

The present experiment used various kinds of stimuli which were not controlled for color, brightness, or content. Thus, some stimuli may have been more salient than others based on these low-level features. As is the case for humans, chimpanzees' visual search patterns are controlled by salient "pop-out" features (*Tomonaga, 1993*; *Tomonaga, 2001*). To evaluate the effect of this phenomenon, we prepared two types of control tests. If degree of salience controlled choices, one would hypothesize that (1) chimpanzees would choose the same stimulus they had chosen previously when only the two choice stimuli presented during the "what-did-you-choose" test trials were presented again and that (2) chimpanzees would choose the same stimulus they had chosen previously when the visual search display that had appeared during the test trials was presented again. If the chimpanzees' behavior were not governed simply by these non-memory stimulus factors, they would choose randomly in the control test trials.

In the first type of control test trial, called the no-sample two-choice test, only two stimuli were presented (Fig. 2F). These two stimuli were identical to those that appeared in the choice phase of the "what-did-you-choose?" test trials (with both homogeneous- and heterogeneous-distractor visual search displays) given to each chimpanzee. If the choice response in the test trial were controlled by the chimpanzees' stimulus preference, they would choose the same stimulus selected during the test trials. In the other control test, we presented the same search display that had been presented in the previous "what-did-you-choose" test trials with homogeneous and heterogeneous distractors, but in the form of a visual search trial, to determine whether the chimpanzees chose the stimulus based on their own stimulus preferences under the heterogeneous-distractor condition. If this were the case, the chimpanzees would choose the same stimulus when the same search display was presented. We administered 32 additional test sessions to each chimpanzee. Each session consisted of two matching trials, 20 visual search trials (12 with homogeneous distractors and eight with heterogeneous distractors), two no-sample two-choice test trials, and two visual search test trials. Test trials were presented randomly during a session, and no food reward or feedback was given after the choices in the test trials.

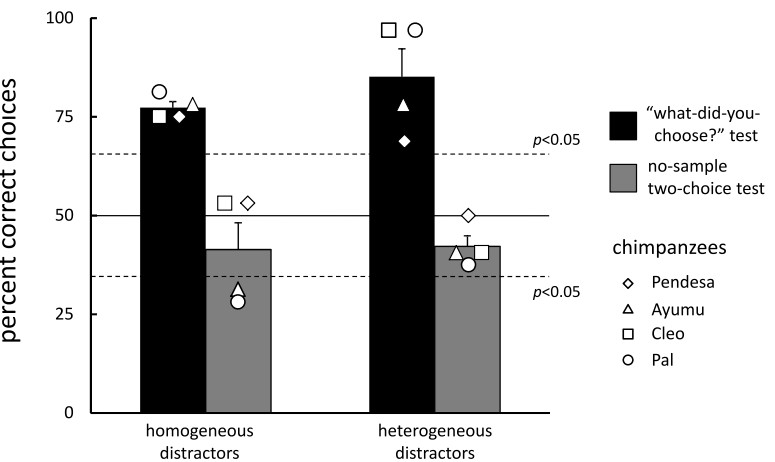

**Figure 3 Mean percentage of correct choices during the "what-did-you-choose?" tests and the no-sample control tests.** Error bars indicate the standard errors. Data from each chimpanzee are also shown.

## RESULTS

All data from each chimpanzee are presented in Dataset S1. All chimpanzees completed the experimental sessions (258 sessions on average, range: 255–260) within 578 days on average (range: 481–664); test trials were given every 9 days. Participants performed very accurately on matching (97.1%, standard error of the mean, $SEM = 0.7$) and homogeneous visual search (94.0%, $SEM = 1.3$) trials. The self-paced delay interval lasted 1.05 s, on average, across all types of trials. Figure 3 (black bars) shows the results of the test trials. All chimpanzees chose the stimulus that was chosen immediately before significantly more often than they chose the other stimulus under both distractor conditions (the significance level at 0.05 was 65.6%, binomial test): 77.4% ($SEM = 1.5$) for the homogeneous- and 85.2% ($SEM = 7.0$) for the heterogeneous-distractor test trials, respectively.

Because the current experiments were conducted over a long period of time, it is possible that performance on the test trials improved during the course of the experimental sessions. To evaluate these learning effects, we divided the 32 test trials under each distractor condition into four eight-trial blocks and compared the performance across blocks. As shown in Table 1, no systematic improvement was observed across blocks. We found a significant difference in Cleo's performance under the homogeneous-distractor condition ($p = 0.038$, Fisher's exact probability test); however, when the data were analyzed based on the first 16 vs. the second 16 trials (68.8% vs. 81.25%), this difference was not significant ($p = 0.685$).

During the control test sessions, chimpanzees performed very accurately on both the matching (98.4%, $SEM = 1.1$) and visual search trials (94.0%, $SEM = 1.7$). For the first type of test trials (i.e., the no-sample two-choice tests), the stimulus choice made by the chimpanzees was random in comparison with their previous choices [41.4% ($SEM = 6.8$) in the homogeneous-display control and 42.2% ($SEM = 2.7$) in the heterogeneous-display control test trials; see Fig. 3, gray bars]. Actually, the performance of two chimpanzees

**Table 1 Learning effects in the "what-did-you-choose" test trials.** Each number represents the number of correct trials in each eight-trial block.

| Eight-trial block | Pendesa homo DSTs | Pendesa hetero DSTs | Ayumu homo DSTs | Ayumu hetero DSTs | Cleo homo DSTs | Cleo hetero DSTs | Pal homo DSTs | Pal hetero DSTs |
|---|---|---|---|---|---|---|---|---|
| 1 | 5 | 5 | 8 | 7 | 3 | 8 | 8 | 8 |
| 2 | 6 | 6 | 6 | 7 | 8 | 8 | 5 | 8 |
| 3 | 7 | 6 | 4 | 5 | 7 | 7 | 5 | 7 |
| 4 | 6 | 5 | 7 | 6 | 6 | 8 | 8 | 8 |
| Fisher's Exact $p$ | 0.942 | 1.000 | 0.145 | 0.791 | 0.038 | 1.000 | 0.053 | 1.000 |

**Notes.**
homoDSTs, trials with homogeneous distractors; heteroDSTs, trials with heterogeneous distractors.

**Table 2 Results of the second control test sessions.** The percentages of correct trials under baseline conditions and the percentage of trials in which chimpanzees chose the same stimuli they had chosen previously in the repeated visual search test trials.

| Chimpanzee | Baseline MTS | Baseline Visual search (homoDSTs) | Test Visual search homoDSTs | Test Visual search heteroDSTs |
|---|---|---|---|---|
| Pendesa | 98.4 | 94.5 | 96.9 | 6.3 |
| Ayumu | 100 | 98.4 | 96.9 | 31.3 |
| Cleo | 95.3 | 90.4 | 87.5 | 21.9 |
| Pal | 100 | 92.7 | 87.5 | 25 |
| Average | 98.4 | 94 | 92.2 | 21.1 |
| SEM | 1.1 | 1.7 | 2.7 | 5.3 |

**Notes.**
homoDSTs, trials with homogeneous distractors; heteroDSTs, trials with heterogeneous distractors; SEM, standard errors of mean.

(Ayumu and Pal) significantly differed from chance in the homogeneous-display control two-choice test trials, but they avoided the stimulus that they had previously chosen. In the other control test of the repeated visual search test trials (Table 2), the chimpanzees chose the same stimulus they had chosen in the previous homogeneous-distractor test trial in 92.2% ($SEM = 2.7$) of the trials. However, they chose the same stimulus as in the previous heterogeneous-distractor test trial in only 21.1% ($SEM = 5.3$) of the trials. Although one chimpanzee exhibited significantly better performance (Ayumu, 31.3%, $p < 0.05$, binomial test) than would be expected by chance (16.7%), the other three chimpanzees chose stimuli randomly.

## DISCUSSION

Irrespective of the type of distractors, all chimpanzees exhibited significantly better recognition of the stimulus chosen immediately before testing than would be expected by chance. These results indicate short-term retention of memories of their own behavior. Furthermore, they performed at the level of chance on control test trials. These data

suggest that the results of "what-did-you-choose" test trials cannot be explained by only the salience of the stimuli.

Our results clearly show that chimpanzees can remember choices they made immediately before an incidental memory task. Unlike the pigeons studied in previous research (*Singer & Zentall, 2007*; *Zentall et al., 2001*; *Zentall, Singer & Stagner, 2008*), the chimpanzees in our study experienced a very long interval (one trial/9 days) between test trials. In contrast with the situation involving pigeons, this long inter-test interval may have made it less likely that the chimpanzees expected to be asked, "what did you choose?" immediately after an individual trial. Thus, the present results do not support rule learning or semantic memory (*Singer & Zentall, 2007*). Their choice behavior during the test trials cannot be explained by simple association learning on the basis of differential reinforcement because the choice of the stimulus on the search display and the subsequent choice between the two stimuli were not actually differentially reinforced by a food or sound reward. Additionally, it would be difficult to explain our results in terms of non-memory strategies such as stimulus preference. The choices of the chimpanzees during the heterogeneous-distractor trials and the choice phase during the test trials were not consistent enough to explain our results. In the test trials, one of the two choice stimuli was selected from the stimuli in the previous search display. This may have led to different degrees of familiarity with the two choice stimuli, and the chimpanzees may have utilized these differences in the choice of a stimulus. This possibility should be examined further in the near future. Of course, it is still rather difficult to distinguish between semantic memory and episodic recollection using the current experimental procedure. It remains plausible that the chimpanzees chose the stimuli based on semantic memory ("that is the stimulus I chose") rather than based on episodic recollection ("I remember choosing that stimulus"). In the future, we should devise a new procedure to differentiate these possibilities more clearly.

Previous studies have reported that chimpanzees have the ability to form long-term episodic-like memories (e.g., *Menzel, 1999*). In our experiments, the delay interval between the choice behavior and the incidental question was only 1 s. Can our findings be connected with the previous results showing long-term retention of memories of their own behavior? Some researchers may argue for a critical gap between the short- and long-term properties of episodic memory, whereas others may not make this argument (*Singer & Zentall, 2007*). We think that our results can be closely related to the "episodic buffer" in the components of working memory proposed by *Baddeley (2000)*. This component is considered to act as a bridge between working memory and long-term episodic memory. Chimpanzees may temporally store memories of their own behavioral episodes in this episodic buffer. As Baddeley proposed, the episodic buffer plays a critical role in feeding information into and retrieving information from long-term episodic memory. Thus, we should focus on the temporal dynamics of episodic-like memory and its underlying mechanisms. However, these mechanisms remain unclear from the perspective of comparative cognition. More detailed comparative studies should be conducted in the future.

In our experiment, only the "what" component of short-term episodic-like memory was examined, as our experimental design did not allow us to determine the "when" and "where" components. For example, the "when" component was always fixed: the choice "just before" the current testing phase. Furthermore, the "where" component could not be manipulated in our setting as we always tested the chimpanzees in the same apparatus in the same room. Although controversy concerning the necessary conditions for testing episodic-like memory in nonhuman animals persists (*Zentall, Singer & Stagner, 2008*), we plan to modify the experimental procedure enough to test the "when" and "where" components in a well-controlled setting. These further attempts will tell us more about episodic-like memory in chimpanzees as well as about the evolution of episodic memory.

## CONCLUSION

In the present study, we examined the ability of chimpanzees to recognize what they had just done using matching-to-sample and visual search paradigms. In the memory-test trials, chimpanzees chose the stimulus that they had chosen previously. These results seem to support the operation of short-term episodic-like memory in chimpanzees. The comparative cognition domain has emphasized the long-term aspects of episodic-like memory more than short-term storage of episodes, which may be attributable to the original definition of episodic memory. However, the present results suggest the need for additional comparative cognitive investigations of the temporal dynamics involved in role of episodic-like memory in the transition from short-term to long-term storage.

## ACKNOWLEDGEMENTS

We thank all of the staff of the Language and Intelligence Section and the Centre for Human Evolution Modelling Research of the Primate Research Institute, Kyoto University for their comments, generous support, and daily care of chimpanzees.

### Funding

The current study and the preparation of the manuscript were financially supported by the Grants-in-Aid for Scientific Research from the Japanese Ministry of Education, Culture, Sports, Science, and Technology (MEXT), Japan Society for the Promotion of Science (JSPS) (#19300091, #20002001, #23220006, and #24000001). The funders had no role in study design, data collection and analysis, decision to publish, or preparation of the manuscript.

### Grant Disclosures

The following grant information was disclosed by the authors:
Japanese Ministry of Education, Culture, Sports, Science, and Technology (MEXT).
Japan Society for the Promotion of Science (JSPS): #19300091, #20002001, #23220006, #24000001.

## Competing Interests

Takaaki Kaneko is an employee of RIKEN Brain Science Institute.

## Author Contributions

- Masaki Tomonaga conceived and designed the experiments, performed the experiments, analyzed the data, contributed reagents/materials/analysis tools, wrote the paper, prepared figures and/or tables, reviewed drafts of the paper.
- Takaaki Kaneko performed the experiments, analyzed the data, reviewed drafts of the paper.

## Animal Ethics

The following information was supplied relating to ethical approvals (i.e., approving body and any reference numbers):

The Animal Welfare and Animal Care Committee of KUPRI, and the Animal Research Committee of Kyoto University: 07-1554.

## Supplemental Information

Supplemental information for this article can be found online at http://dx.doi.org/10.7717/peerj.637#supplemental-information.

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
