# Peer review of "What did you choose just now? Chimpanzees’ short-term retention of memories of their own behavior"

_PeerJ, doi:10.7717/peerj.637_

## Round 0.1 · original submission · Major Revisions

The paper has been judged as interesting by all the reviewers. However, several serious concerns have been raised. I therefore suggest to analytically answer these points and to seriously improve the quality of writing. A final reading by an English native speaker could be useful.

·

Basic reporting

In this study, the authors demonstrated the ability of episodic memory in chimpanzees. The chimpanzees were unexpectedly required to choose one of two test stimuli, and exhibited behavior based on their own previous choice behavior.

Experimental design

The procedure of the test trial was elaborated.

Validity of the findings

The performance of the chimpanzees in the test was satisfactory.

Additional comments

I suggest that the authors discuss more about the significance of their findings, i.e., episodic-like memory in chimpanzees. The authors might want to discuss it from ’the perspective of comparative cognition’ as they mentioned in the text.


The following is the minor points.

Line 64: The numbers should be filled.

Line 112: The numbers of trials look odd. The authors must check the numbers.

Line 134: In the second control test, the performance of 3 of 4 chimpanzees exceeded the chance level even though they were not statistically significant. The authors' interpretation, the 3 chimpanzees chose stimuli randomly, seems to be dangerous a little.

Line 169: The meaning of 'integrated' is not clear. Clear description is expected.

Line 179: The contents of the second paragraph of the "Conclusion" section should be discussed in the "Discussion" section.

Line 231: In the Table 1, the meaning of the numbers in the "Baseline" columns were different from those in the "Test" columns. The difference should be mentioned.

Reviewer 2 ·

Basic reporting

The basic reporting is generally acceptable, but I suggest the following changes to help improve the paper. These are listed below with reference to line number where possible.

This study could be better placed in the context of other comparative work if the authors mention somewhere that the unexpected question paradigm has also been used in rats (e.g., Zhou et al., 2012).

To avoid misleading readers, the authors should tone down their claims that many species show episodic memory (line 9). There is considerable debate about what evidence is actually needed to infer human-like episodic memory in nonhumans (e.g., Tulving, 2005; Suddendorf and Busby, 2003). Mentioning these dissenting opinions will strengthen the paper and help place it better in the context of existing literature.

An optional comment: Given the previous about the difficulty of testing human-like episodic memory in nonhumans, the authors might strengthen the introduction of their paper by admitting up-front that they cannot actually test episodic memory in the way it is claimed for humans. However, they might then motivate their study by noting that episodic memory has many attributes that are testable in nonhumans (e.g., it’s recollective, it often codes the order of events, the context, etc), that one of those aspects is the ability to retroactively retrieve incidentally-encoded information, and that this can be tested in nonhumans. The claim that they are testing one aspect of episodic memory would be much more defensible and fair to the literature.

The authors introduce the what/when/where criteria (lines 7-8) and then immediately mention four studies that claim to “show such episodic memory” (lines 11). This will probably be understood by most readers as claiming that these studies show evidence for what/when/where memory. Clayton et al. (1998) certainly does, so no problem there. Hampton et al. (2005) actually found little evidence for memory of the “when” component in rhesus monkeys, as indicated by their title. Marcado et al (1998) used a repeat command in dolphins, which is very similar to the unexpected question paradigm and should certainly be covered in the introduction, but is not a test of what/when/where. Crystal and Babb (2008) tested spatial memory after long delays, with no test of the what or when components. To avoid misleading readers, this coverage of the previous literature should be reworked.

In their initial introduction of their test, the authors mention that chimpanzees were “presented with red and green choices” (line 35). This is false (see Figure 2) and should probably be changed to avoid confusion.

The authors should report the sex and age of their four subjects (lines 42-47).

The measurements are missing for the blue square (line 64) and green circle (line 67).

“Touching the choice stimulus, which was the same as the sample, delivered food…” might be better rephrased as “Touching the choice stimulus that was the same as the sample delivered food…” (line 69).

The methods might be clearer if the authors reported what constituted an error (line 70).

“As for the matching trials…” might be better rephrased as “As in the matching trials…” (line 73).

The sentence that begins with “Touching the stimulus…” needs to be rephrased to make it clearer what actually happened (lines 78-79).

Was the chosen stimulus in the visual search task re-presented in the “what-did-you-choose” test only if the chimps chose correctly? Or was it also re-presented if the chimps chose incorrectly in the homogeneous distractor test? (lines 83-84).

“Any touch to the stimulus” should probably be rephrased as “Any touch to a stimulus” or “Any touch to the stimuli” (line 84).

“as well as humans” should probably be rephrased as “as in humans” or “like humans” (line 94).

Predictions 1 and 2 need to be edited for grammar and tense agreement (lines 96-100).

The authors should double-check the reported numbers for the control experiments. They report “20 visual search trials (12 with homogeneous distractors and 12 with heterogeneous distractors)” but that doesn’t add up (lines 111-112).

To avoid seeming biased, the authors should probably remove the phrase “To rule out this possibility” (line 95). Perhaps replace with “To evaluate this possibility”.

The words “the evidence for the” could be removed from the sentence (line 140).

It could be clearer what the authors mean by making the distinction between the “temporal dynamics of episodic-like memory” and the “distinctions between short- versus long-term memory” (lines 170-171). Is memory duration/decay not a temporal dynamic?

Figure 2 shows the green dot appearing in random locations. If this is the case, it should probably be mentioned in text.

The trial types in Figure 2 might be made clearer if “matching trial” were moved up to be on the same level as “visual search trial” and ““what—did-you-choose?” test trial”. Also, the no-sample two-choice test might be better with a comparable title, like “control test”

Figure 2 almost looks like the test location was the same as during study. I assume this was not the case, so it might be best to display it in a different location to avoid confusing readers.

Experimental design

The experimental design is clean and straight-forward. The sparse distribution of test trials is an especially nice feature, as the authors point out.

The only small blemish I spotted was that the test trials in the control conditions should have been placed at the end of the session so they were better matched with the probe trials in the main experiment (or the probe trials in the main experiment should have been presented at random). Also, the composition of the control sessions (e.g., number of matching trials, number of visual search trials, etc) should have been the same as in the main experiment. But this is a minor problem and this inconsistency would be unlikely to have produced the reported results.

In text, it’s a little unclear whether the chimps received a reward for doing the visual search portion of the “what-did-you-choose?” test. Based on Figure 2, I assume this was not the case. But the authors should make this very clear in text. Otherwise, if the chimps were indeed choosing between a previously-rewarded item and a not-previously-rewarded item in the “what-did-you-see?” tests, that would be a huge confound and would require a more major revision.

Validity of the findings

Based on the methods, the actual behavioral results seem valid. I do not doubt that the chimpanzees remembered what they had chosen in the visual search trials.

The interpretation is generally good. For the most part, the authors make clear what has been demonstrated (short-term memory for a visual stimulus when tested in an unexpected manner) and what is speculation (episodic memory, perhaps like Baddeley’s episodic buffer).

I disagree slightly with the strong statement that “the present results do not support rule learning or semantic memory” (lines 147-148). As the authors point out later, it’s possible that the chimps simply found the selected stimulus more familiar during the probe tests (lines 154-157), that this sense of familiarity was weak due to the limited encoding, and that it had faded by the time control tests were run. It’s also possible that chimps had a semantic memory of the stimulus rather than an episodic recollection of having personally chosen it (e.g., “that’s the chosen stimulus” vs. “I remember choosing that stimulus”). Without the ability to ask them if they remember the item or just know they saw it, it’s difficult to know one way or the other. Perhaps the authors might make this discussion stronger by being more honest that it’s difficult to know for sure exactly how chimps are completing this task. They can then still speculate that chimps are using episodic or an episodic-like memory, as long as they are clear about what has been demonstrated and what must be speculated.

Additional comments

This was a nice study. I enjoyed reading it.

Reviewer 3 ·

Basic reporting

Overall, the topic of the study is very interesting since, compared to rodents and birds, nonhuman primates have received relatively little attention in episodic-like memory research. For this reason, I also suggest to mentioned in the manuscript other studies on episodic-like memory in nonhuman primates which have been neglected in the present version of the manuscript, as for example a study by Martín-Ordás et al. [Martín-Ordás, G., Haun, D., Colmenares, F. & Call, J. 2010. Keeping track of time: evidence of episodic-like memory in great apes. Animal Cognition. 13, 331-340] and a study by Hoffman et al. [Hoffman ML, Beran, MJ, Washburn DA. 2009. Memory for ‘‘what’’, ‘‘where’’ and ‘‘when’’ information in rhesus monkeys (Macaca mulatta). J Exp Psychol Anim Behav Proc, 35, 143-152]. In this way, the authors can also highlight how their work fits into this field of knowledge.

In my opinion, the authors should better clarify two aspects that they considered only marginally in the introduction of their study:
1. First, since episodic memory is defined as a long-term memory system whereas here they chose to carry out a study involving short-term memory, it would be better to highlight which is the relevance in doing so and how it can contribute to improve our knowledge about mechanisms involved in the episodic memory. Unfortunately, this aspect was considered only in the last paragraph of the discussion.
2. Moreover, since episodic memory in animal species is usually referred to as “episodic-like” memory rather than “episodic” memory because it does not evaluate subjective experiences that are usually associated to conscious recollection in humans (e.g., Clayton and Dickinson 1998, Clayton et al. 2003), more attention should be devoted to explain it. This could improve the clarity of the topic under study also for non-specialists in memory in nonhuman species.

Experimental design

The research question and how it has been implemented in terms of experimental design should be better clarified. It is not easy to understand how the complex experimental design adopted, which includes several conditions, answers to the research question. I think that a greater effort in this direction could really improve the manuscript. I found the methods quite confusing and had to read them through several times before I finally understood them. Therefore, the methods could use some clarification (see some details below).

Line 42: Please, add information about sex and age of the chimpanzees who participated to the experiments.

Lines 42-43: Why figure 1 has been cited here? I suggest to move the reference elsewhere (for example, in the first paragraph of the “Apparatus and stimuli” section).

Lines 42-47: It is not specified if chimpanzees were alone during the experimental sessions. According to the dimension of the boot (see description of the experimental setting, line 54), one can guess so, however, in Figure 1, it is possible to see a second chimpanzee behind the individual in the foreground.

Line 57: “Food reward was delivered…by the universal feeder”. If “universal feeder” is the name of a specific model of feeder, put the name in brackets or capitalize the name, otherwise, if it is the name for a class of feeders it would be better to write “by a universal feeder”.

Lines 57-58: Did the authors use a commercial software to program data collection by PC or they use a particular kind of programming language? Please specify.

Line 64: Specify the correct size of the blue square.

Line 67: Specify the correct size the green circle.

Lines 72-79: Whereas the homogeneous distractor condition is a visual search problem (search the figure which differs from the other figures), the heterogeneous distractor condition it seems to me that it is not since no visual search behavior is required (subjects simply had to touch one figure, but it could be any figure).

Lines 78-79: Specify also in the text that chimpanzees obtained the reward after touching the green circle (now is understandable only by the schema depicted in figure 2).

Lines 81-82: Reading the description of the matching to sample task I have had the impression that the “self-paced delay interval” is referred only to the delay interval between the disappearance of the sample stimulus and the appearance of the comparison stimuli, which was under control of the subject by touching the green circle. Here I realized that they use this definition also to refer to the interval between the disappearance of the stimuli and the attainment of the reward in the visual search task. It should be better to clarify this in the description of the visual search task.

Lines 81-84: In the “what-did-you-choose” test two choice stimuli occasionally appeared after the visual search task. One stimulus was the one the chimpanzee chose from the search display, and the other one was a stimulus which differed from any stimulus in the previous search display. Any touch to the stimulus yielded no feedback. It is not clear to me how the author can rule out the possibility that the chimpanzee could choose the stimulus on the basis of the fact that they were probably be rewarded for the same stimulus shortly before. This behavior could be driven by memory mechanisms which do not necessarily involve “episodic” memory in my opinion.

Lines 85-90 and 111-114: It is not clear how many sessions/trials subjects received on a daily basis. Please specify how the total number of session/trials have been distributed over the days.

Line 102: “In the first type of test trial…” should be “In the first type of control test trial…” To avoid to be confounding, the same would be any time the authors mention the control test.
Lines 107-109: It is not clear how exactly worked the second type of control test presented.

Validity of the findings

The Authors carried out a series of experimental conditions aimed to assess the effect of the short-term episodic-like memory on chimpanzees. They also did control tests to rule out the effect of non-memory factors due to different salience of the stimuli (e.g., different perceptual features).
In my opinion the experimental design they adopted is convincing in demonstrating that the choice behavior rely on short-term memory cues, however it is less clear why episodic memory should be involved in the solution of this kind of task. The most simple explanation which takes into account a similar pattern of results is that subjects could simply choose a certain stimulus since that stimulus has been recently rewarded in a previous choice task. This requires short-term memory abilities that do not necessarily involve episodic-like memory, i.e., the ability to recall past personal particular experiences (what) that occurred at a particular time (when) and place (where). Even if in the conclusion section the authors admit that the only component they tested was the “what” component, the episodic memory is intrinsically defined by the coexistence of all these three components. Therefore, unless the authors have stronger argumentations I suggest to revise part of the discussion and conclusion to further downscale explanations related to the episodic memory.

It would be of interest to know if there were changes in the performance of the “what-did-you-choose” test through the time. Especially because chimpanzees employed long time to complete the experimental sessions (578 days on average). Did the authors assessed effect due to the practice with the task?

I appreciated that the authors present data results at individual level, however since they also show mean values for the group, I suggest to insert statistical analyses at group level as well (one possibility, if applicable, could be to use a one-sample t test to evaluate whether the mean percentage of the group differed significantly from a chance distribution with a mean of 50%).

The results showed that “All chimpanzees chose the stimulus that was chosen immediately before significantly more often than they chose the other stimulus under both distractor conditions (significance level at 0.05 was 65.6%, binomial test): 77.4% (SE = 1.5) for homogeneous and 85.2% (SE = 7.0) for the heterogeneous distractor test trials, respectively”. Did the authors assessed if chimpanzees were significantly more accurate in recognize the stimulus under the heterogeneous distracter condition compared to the homogeneous one? This could highlight a possible effect of the choice condition on memory mechanisms, since in the homogeneous condition chimpanzees had to choose according to a rule (choose the stimulus which differ from the other stimuli) whereas in the heterogeneous condition they could freely choose any of the stimuli presented.

Lines 122-123: Did the authors apply the binomial test with approximation to the normal distribution (i.e., binomial z-scores)? Please specify.

Lines 129-131: “Actually, two chimpanzees (Ayumu and Pal) showed performances significantly different from the chance level in homogeneous-display control two-choice test trials, but they “avoided” the stimulus which was chosen previously by themselves.” Do the authors have any explanation for the behavior of these two chimpanzees in this particular condition?

---

## Round 0.2 · Minor Revisions

The reviewers have appreciated your effort. Please address these remaining minor issues.

Reviewer 3 ·

Basic reporting

No comments

Experimental design

No comments

Validity of the findings

No comments

Additional comments

The Authors considered the suggestions I provided for the first review of the paper. I appreciated the revised version of the manuscript. Here are some further minor revisions:

Line 13. Delete the second bracket just after ‘(Mercado III et al., 1998)’

Since it has been used the standard error of the mean, it would be better to use the abbreviation ‘SEM’ instead of ‘SE’.

Lines 159-160. It would be better to avoid using round brackets within other round brackets. Therefore, add a comma after ‘standard error of the mean’ and delete the brackets from ‘(SE)’.

Lines 176-177. In this case, square brackets can be used to avoid using round brackets within other round brackets: [41.4% (SE = 6.8) in the homogeneous-display control and 42.2% (SE=2.7) in the heterogeneous display control test trials; see Fig. 3, gray bars]

Lines 199-207. ‘Additionally, it would be difficult to explain our results in terms of non-memory strategies such as stimulus preference. The choices of the chimpanzees during the heterogeneous-distractor trials and the choice phase during the test trials were not consistent enough to explain our results. In the test trials, one of the two choice stimuli was selected from the stimuli in the previous search display. This may have led to different degrees of familiarity with the two choice stimuli, and the chimpanzees may have utilized these differences in the choice of a stimulus. This possibility should be examined further in the near future. The results of the additional tests, however, may rule out this possibility. The chimpanzees did not show a significant preference for the stimulus that was chosen previously and with which they were more familiar even though this familiarity drew on their long-term memory.’
This part of the discussion is confusing, in particular because the two sentences here reported in bold seem to be in contradiction. Please clarify.

Lines 210-211. Add ‘on’ between ‘based’ and ‘semantic memory’

Line 215. Since the authors quoted only the study of Menzel (1968), they should add also ‘e.g.,’.

---

## Round 0.3 · accepted · Accept

The manuscript has now improved a lot and can be accepted for publication